# PERSONALIZED REWARD LEARNING WITH INTERACTION-GROUNDED LEARNING (IGL)

**Jessica Maghakian**
Stony Brook University
jessica.maghakian@stonybrook.edu

**Paul Mineiro**
Microsoft Research NYC
pmineiro@microsoft.com

**Kishan Panaganti**
Texas A&M University
kpb@tamu.edu

**Mark Rucker**
University of Virginia
mr2an@virginia.edu

**Akanksha Saran**
Microsoft Research NYC
akanksha.saran@microsoft.com

**Cheng Tan**
Microsoft Research NYC
tan.cheng@microsoft.com

## ABSTRACT

In an era of countless content offerings, recommender systems alleviate information overload by providing users with personalized content suggestions. Due to the scarcity of explicit user feedback, modern recommender systems typically optimize for the same fixed combination of implicit feedback signals across all users. However, this approach disregards a growing body of work highlighting that (i) implicit signals can be used by users in diverse ways, signaling anything from satisfaction to active dislike, and (ii) different users communicate preferences in different ways. We propose applying the recent Interaction Grounded Learning (IGL) paradigm to address the challenge of learning representations of diverse user communication modalities. Rather than requiring a fixed, human-designed reward function, IGL is able to learn personalized reward functions for different users and then optimize directly for the latent user satisfaction. We demonstrate the success of IGL with experiments using simulations as well as with real-world production traces.

## 1 INTRODUCTION

From shopping to reading the news, modern Internet users have access to an overwhelming amount of content and choices from online services. Recommender systems offer a way to improve user experience and decrease information overload by providing a customized selection of content. A key challenge for recommender systems is the rarity of explicit user feedback, such as ratings or likes/dislikes (Grčar et al., 2005). Rather than explicit feedback, practitioners typically use more readily available implicit signals, such as clicks (Hu et al., 2008), webpage dwell time (Yi et al., 2014), or inter-arrival times (Wu et al., 2017) as a proxy signal for user satisfaction. These implicit signals are used as the reward objective in recommender systems, with the popular Click-Through Rate (CTR) metric as the gold standard for the field (Silveira et al., 2019). However, directly using implicit signals as the reward function presents several issues.

*Implicit signals do not directly map to user satisfaction.* Although clicks are routinely equated with user satisfaction, there are examples of unsatisfied users interacting with content via clicks. Clickbait exploits cognitive biases such as caption bias (Hofmann et al., 2012) or the curiosity gap (Scott, 2021) so that low quality content attracts more clicks. Direct optimization of the CTR degrades user experience by promoting clickbait items (Wang et al., 2021). Recent work shows that users will even click on content that they know a priori they will dislike. In a study of online news reading, Lu et al. (2018a) discovered that 15% of the time, users would click on articles that they strongly disliked. Similarly, although longer webpage dwell times are associated with satisfied users, a study by Kim et al. (2014) found that dwell time is also significantly impacted by page topic, readability and content length.

*Different users communicate in different ways.* Demographic background is known to have an impact on the ways in which users engage with recommender systems. A study by Beel et al. (2013) shows that older users have CTR more than 3x higher than their younger counterparts. Gender also has an impact on interactions, e.g. men are more likely to leave dislikes on YouTube videos than women (Khan, 2017). At the same time, a growing body of work shows that recommender systems do not provide consistent performance across demographic subgroups. For example, multiple studies on ML fairness in recommender systems show that women on average receive less accurate recommendations compared to men (Ekstrand et al., 2018; Mansoury et al., 2020). Current systems are also unfair across different age brackets, with statistically significant recommendation utility degradation as the age of the user increases (Neophytou et al., 2022). The work of Neophytou et al. identifies usage features as the most predictive of mean recommender utility, hinting that the inconsistent performance in recommendation algorithms across subgroups arises from differences in how users interact with the recommender system.

These challenges motivate the need for personalized reward functions. However, extensively modeling the ways in which implicit signals are used or how demographics impact interaction style is costly and inefficient. Current state-of-the-art systems utilize reward functions that are manually engineered combinations of implicit signals, typically refined through laborious trial and error methods. Yet as recommender systems and their users evolve, so do the ways in which users implicitly communicate preferences. Any extensive models or hand tuned reward functions developed now could easily become obsolete within a few years time.

To this end, we propose Interaction Grounded Learning (IGL) Xie et al. (2021) for personalized reward learning (`IGL-P`). IGL is a learning paradigm where a learner optimizes for unobservable rewards by interacting with the environment and associating observable feedback with the true latent reward. Prior IGL approaches assume the feedback either depends on the reward alone (Xie et al., 2021), or on the reward and action (Xie et al., 2022). These methods are unable to disambiguate personalized feedback that depends on the context. Other approaches such as reinforcement learning and traditional contextual bandits suffer from the choice of reward function. However our proposed personalized IGL, `IGL-P`, resolves the 2 above challenges while making minimal assumptions about the value of observed user feedback. Our new approach is able to incorporate both explicit and implicit signals, leverage ambiguous user feedback and adapt to the different ways in which users interact with the system.

**Our Contributions:** We present `IGL-P`, the first IGL strategy for context-dependent feedback, the first use of inverse kinematics as an IGL objective, and the first IGL strategy for more than two latent states. Our proposed approach provides an alternative to agent learning methods which require handcrafted reward functions. Using simulations and real production data, we demonstrate that `IGL-P` is able to learn personalized rewards when applied to the domain of online recommender systems, which require at least 3 reward states.

## 2 PROBLEM SETTING

### 2.1 CONTEXTUAL BANDITS

The contextual bandit (Auer et al., 2002; Langford & Zhang, 2007) is a statistical model of myopic decision making which is pervasively applied in recommendation systems (Bouneffouf et al., 2020). IGL operates via reduction to contextual bandits, hence, we briefly review contextual bandits here.

The contextual bandit problem proceeds over $T$ rounds. At each round $t \in [T]$, the learner receives a context $x_t \in \mathcal{X}$ (the context space), selects an action $a_t \in \mathcal{A}$ (the action space), and then observes a reward $r_t(a_t)$, where $r_t : \mathcal{A} \to [0, 1]$ is the underlying reward function. We assume that for each round $t$, conditioned on $x_t$, $r_t$ is sampled from a distribution $\mathbb{P}_{r_t}(\cdot \mid x_t)$. A contextual bandit (CB) algorithm attempts to minimize the cumulative regret

$$\mathbf{Reg}_{\mathsf{CB}}(T) := \sum_{t=1}^{T} r_t(\pi^\star(x_t)) - r_t(a_t) \tag{1}$$

relative to an optimal policy $\pi^\star$ over a policy class $\Pi$.

In general, both the contexts $x_1, \ldots, x_T$ and the distributions $\mathbb{P}_{r_1}, \ldots, \mathbb{P}_{r_T}$ can be selected in an arbitrary, potentially adaptive fashion based on the history. In the sequel we will describe IGL in a stochastic environment, but the reduction induces a nonstationary contextual bandit problem, and therefore the existence of adversarial contextual bandit algorithms is relevant.

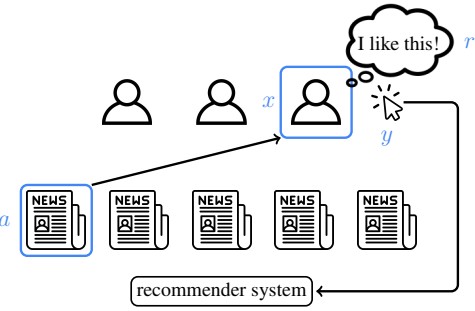

Figure 1: IGL in the recommender system setting. The learner observes the context $x$, plays an action $a$, and then observes a feedback $y$ (that is dependent on the latent reward $r$), but not $r$ itself.

## 2.2 INTERACTION GROUNDED LEARNING

IGL is a problem setting in which the learner's goal is to optimally interact with the environment with no explicit reward to ground its policies. IGL extends the contextual bandit framework by eliding the reward from the learning algorithm and providing feedback instead (Xie et al., 2021). We describe the stochastic setting where $(x_t, r_t, y_t) \sim D$ triples are sampled iid from an unknown distribution; the learner receives the context $x_t \in \mathcal{X}$, selects an action $a_t \in \mathcal{A}$, and then observes the feedback $y_t(a_t)$, where $y_t : \mathcal{A} \rightarrow [0, 1]$ is the underlying feedback function. Note $r_t(a_t)$ is never revealed to the algorithm: nonetheless, the regret notion remains the same as Eq. (1). An information-theoretic argument proves assumptions relating the feedback to the underlying reward are necessary to succeed (Xie et al., 2022).

### 2.2.1 SPECIALIZATION TO RECOMMENDATION

For specific application in the recommendation domain, we depart from prior art in IGL (Xie et al., 2021; 2022) in two ways: first, in the assumed relationship between feedback and underlying reward; and second, in the number of latent reward states. For the remainder of the paper, we represent the users as $x$ (i.e., as context), content items as action $a$, the user satisfaction with recommended content as $r$, and the feedback signals in response to the recommended content (user interactions with the system interface) as $y$.

**Feedback Dependence Assumption**  Xie et al. (2021) assumed full contextual independence of feedback on context and chosen action, i.e. $y \perp x, a | r$. For recommender systems, this implies that all users communicate preferences identically for all content. In a subsequent paper, Xie et al. (2022) loosen the full conditional independence by considering context conditional independence, i.e. $y \perp x | a, r$. For our setting, this corresponds to the user feedback varying for combinations of preference and content, but remaining consistent across all users. Neither of these two assumptions are natural in the recommendation setting because different users interact with recommender systems in different ways (Beel et al., 2013; Shin, 2020). In this work, we assume $y \perp a | x, r$, i.e., the feedback $y$ is independent of the displayed content $a$ given the user $x$ and their disposition toward the displayed content $r$. Thus, we assume that users may communicate in different ways, but a given user expresses satisfaction, dissatisfaction and indifference to all content in the same way.

**Number of Latent Reward States**  Prior work demonstrates that a binary latent reward assumption, along with an assumption that rewards are rare under a known reference policy, are sufficient for IGL to succeed. Specifically, optimizing the contrast between a learned policy and the oblivious uniform policy is able to succeed when feedback is both context and action independent (Xie et al., 2021); and optimizing the contrast between the learned policy and all constant-action policies succeeds when the feedback is context independent (Xie et al., 2022).

Although the binary latent reward assumption (e.g., satisfied or dissatisfied) appears reasonable for recommendation scenarios, it fails to account for user indifference versus user dissatisfaction. This observation was first motivated by our production data, where a 2 state IGL policy would sometimes maximize feedback signals with obviously negative semantics. Assuming users ignore most content most of the time (Nguyen et al., 2014), negative feedback can be as difficult to elicit as positive feedback, and a 2 state IGL model is unable to distinguish between these extremes. Hence, we posit a minimal latent state model for recommender systems involves 3 states: (i) $r = 1$, when users are satisfied with the recommended content, (ii) $r = 0$, when users are indifferent or inattentive, and (iii) $r = -1$, when users are dissatisfied. See Appendix A.1 and Maghakian et al. (2022) for details.

## 3 DERIVATIONS

Prior approaches to IGL use contrastive learning objectives (Xie et al., 2021; 2022), but the novel feedback dependence assumption in the prior section impedes this line of attack. Essentially, given arbitrary dependence upon $x$, learning must operate on each example in isolation without requiring comparison across examples. This motivates attempting to predict the current action from the current context and the currently observed feedback, i.e., inverse kinematics.

**Inverse Kinematics**  Traditionally in robotics and computer animation, inverse kinematics is the mathematical process used to recover the movement (action) of a robot/object in the world from some other data such as the position and orientation of a robot manipulator/video of a moving object (in our case, the other data is context and feedback). We motivate our inverse kinematics strategy using exact expectations. When acting according to any policy $P(a|x)$, we can imagine trying to predict the action taken given the context and feedback; the posterior distribution is

$$P(a|y,x) = \frac{P(a|x)P(y|a,x)}{P(y|x)} \qquad \text{(Bayes rule)}$$

$$= P(a|x) \sum_r \frac{P(y|r,a,x)}{P(y|x)} P(r|a,x) \qquad \text{(Total Probability)}$$

$$= P(a|x) \sum_r \frac{P(y|r,x)}{P(y|x)} P(r|a,x) \qquad (y \perp a|x,r)$$

$$= P(a|x) \sum_r \frac{P(r|y,x)}{P(r|x)} P(r|a,x) \qquad \text{(Bayes rule)}$$

$$= \sum_r P(r|y,x) \left( \frac{P(r|a,x)P(a|x)}{\sum_a P(r|a,x)P(a|x)} \right). \qquad \text{(Total Probability)} \qquad (2)$$

We arrive at the inner product between a reward decoder term ($P(r|y,x)$) and a reward predictor term ($P(r|a,x)$).

**Extreme Event Detection**  Direct extraction of a reward predictor using maximum likelihood on the action prediction problem with Eq. (2) is frustrated by two identifiability issues: first, this expression is invariant to a permutation of the rewards on a context dependent basis; and second, the relative scale of two terms being multiplied is not uniquely determined by their product. To mitigate the first issue, we assume $\sum_a P(r=0|a,x)P(a|x) > \frac{1}{2}$, i.e., nonzero rewards are rare under $P(a|x)$; and to mitigate the second issue, we assume the feedback can be perfectly decoded, i.e., $P(r|y,x) \in \{0,1\}$. Under these assumptions, we have

$$r = 0 \implies P(a|y,x) = \frac{P(r=0|a,x)P(a|x)}{\sum_a P(r=0|a,x)P(a|x)}$$
$$\leq 2P(r=0|a,x)P(a|x) \leq 2P(a|x). \qquad (3)$$

Eq. (3) forms the basis for our extreme event detector: anytime the posterior probability of an action is predicted to be more than twice the prior probability, we deduce $r \neq 0$.

Note a feedback merely being apriori rare or frequent (i.e., the magnitude of $P(y|x)$ under the policy $P(a|x)$) does not imply that observing such feedback will induce an extreme event detection; rather the feedback must have a probability that strongly depends upon which action is taken. Because feedback is assumed conditionally independent of action given the reward, the only way for feedback to help predict which action is played is via the (action dependence of the) latent reward.

**Extreme Event Disambiguation**  With 2 latent states, $r \neq 0 \implies r = 1$, and we can reduce to a standard contextual bandit with inferred rewards $\mathbb{1}(P(a|y,x) > 2P(a|x))$. With 3 latent states, $r \neq 0 \implies r = \pm 1$, and additional information is necessary to disambiguate the extreme events. We assume partial reward information is available via a "definitely negative" function[1] $\text{DN} : \mathcal{X} \times \mathcal{Y} \to \{-1, 0\}$ where $P(\text{DN}(x,y) = 0|r=1) = 1$ and $P(\text{DN}(x,y) = -1|r=-1) > 0$. This reduces extreme event disambiguation to one-sided learning (Bekker & Davis, 2020) applied only to

---

[1]"Definitely positive" information can be incorporated analogously.

extreme events, where we try to predict the underlying latent state given $(x, a)$. We assume partial labelling is selected completely at random Elkan & Noto (2008) and treat the (constant) negative labelling propensity $\alpha$ as a hyperparameter. We arrive at our 3-state reward extractor

$$\rho(x, a, y) = \begin{cases} 0 & P(a|y, x) \leq 2P(a|x) \\ -\alpha^{-1} & P(a|y, x) > 2P(a|x) \text{ and } \mathtt{DN}(x, y) = -1 \\ 1 & \text{otherwise} \end{cases}, \qquad (4)$$

equivalent to Bekker & Davis (2020, Equation 11). Setting $\alpha = 1$ embeds 2-state IGL.

---

**Algorithm 1** IGL; Inverse Kinematics; 2 or 3 Latent States; On or Off-Policy.

---

**Input:** Contextual bandit algorithm `CB-Alg`.
**Input:** Calibrated weighted multiclass classification algorithm `MC-Alg`.
**Input:** Definitely negative oracle `DN`.                          # $\mathtt{DN}(\ldots) = 0$ for 2 state IGL
**Input:** Negative labelling propensity $\alpha$.                   # $\alpha = 1$ for 2 state IGL
**Input:** Action set size $K$.
 1: $\pi \leftarrow$ new `CB-Alg`.
 2: $\mathtt{IK} \leftarrow$ new `MC-Alg`.
 3: **for** $t = 1, 2, \ldots;$ **do**
 4:    Observe context $x_t$ and action set $A_t$ with $|A_t| = K$.
 5:    **if** On-Policy IGL **then**
 6:       $P(\cdot|x_t) \leftarrow \pi.\text{predict}(x_t, A_t)$.
 7:       Play $a_t \sim P(\cdot|x_t)$ and observe feedback $y_t$.
 8:    **else**
 9:       Observe $(x_t, a_t, y_t, P(\cdot|x_t))$.
10:    $w_t \leftarrow 1/(KP(a_t|x_t))$.                             # Synthetic uniform distribution
11:    $\hat{P}(a_t|y_t, x_t) \leftarrow \mathtt{IK}.\text{predict}((x_t, y_t), A_t, a_t)$.    # Predict action probability
12:    **if** $K\hat{P}(a_t|y_t, x_t) \leq 2$ **then**               # $\hat{r}_t = 0$
13:       $\pi.\text{learn}(x_t, a_t, A_t, r_t = 0)$
14:    **else**                                                      # $\hat{r}_t \neq 0$
15:       **if** $\mathtt{DN}(\ldots) = 0$ **then**
16:          $\pi.\text{learn}(x_t, a_t, A_t, r_t = 1, P(\cdot|x_t))$
17:       **else**                                                   # Definitely negative
18:          $\pi.\text{learn}(x_t, a_t, A_t, r_t = -\alpha^{-1}, P(\cdot|x_t))$
19:    $\mathtt{IK}.\text{learn}((x_t, y_t), A_t, a_t, w_t)$.

---

**Implementation Notes**   In practice, $P(a|x)$ is known but the other probabilities are estimated. $\hat{P}(a|y, x)$ is estimated online using maximum likelihood on the problem predicting $a$ from $(x, y)$, i.e., on a data stream of tuples $((x, y), a)$. The current estimates induce $\hat{\rho}(x, a, y)$ based upon the plug-in version of Eq. (4). In this manner, the original data stream of $(x, a, y)$ tuples is transformed into a stream of $(x, a, \hat{r} = \hat{\rho}(x, a, y))$ tuples and reduced to a standard online CB problem.

As an additional complication, although $P(a|x)$ is known, it is typically a good policy under which rewards are not rare (e.g., offline learning with a good historical policy; or acting online according to the policy being learned by the IGL procedure). Therefore, we use importance weighting to synthesize a uniform action distribution $P(a|x)$ from the true action distribution.[2] Ultimately we arrive at the procedure of Algorithm 1.

## 4   EMPIRICAL EVALUATIONS

**Evaluation Settings:** Settings include simulation using a supervised classification dataset, online news recommendation on Facebook, and a production image recommendation scenario.

---

[2]When the number of actions varies from round to round, we use importance weighting to synthesize a non-uniform action distribution with low rewards, but we elide this detail for ease of exposition.

**Abbreviations:** Algorithms are denoted by the following abbreviations: Personalized IGL for 2 latent states (`IGL-P(2)`); Personalized IGL for 3 latent states (`IGL-P(3)`); Contextual Bandits for the Facebook news setting that maximizes for emoji, non-like click-based reactions (`CB-emoji`); Contextual Bandits for the Facebook news setting that maximizes for comment interactions (`CB-comment`); Contextual Bandits for the production recommendation scenario that maximizes for CTR (`CB-Click`).

**General Evaluation Setup:** At each time step $t$, the context $x_t$ is provided from either the simulator (Section 4.1, Section 4.2) or the logged production data (Section 4.3). The learner then selects an action $a_t$ and receives feedback $y_t$. In these evaluations, each user provides feedback in exactly one interaction and different user feedback signals are mutually exclusive, so that $y_t$ is a one-hot vector. In simulated environments, the ground truth reward is sometimes used for evaluation but never revealed to the algorithm.

**Code:** Our code[3] is available for all publicly replicable experiments (i.e. except production data).

### 4.1 COVERTYPE IGL SIMULATION

To highlight that personalized IGL can distinguish between different user communication styles, we create a simulated 2-state IGL scenario from a supervised classification dataset. First, we apply a supervised-to-bandit transform to convert the dataset into a contextual bandit simulation (Bietti et al., 2021), i.e., the algorithm is presented the example features as context, chooses one of the classes as an action, and experiences a binary reward which indicates whether or not it matches the example label. In the IGL simulation, this reward is experienced but not revealed to the algorithm. Instead, the latent reward is converted into a feedback signal as follows: each example is assigned one of $N$ different user ids and the user id is revealed to the algorithm as part of the example features. The simulated user will generate feedback in the form of one of $M$ different word ids. Unknown to the algorithm, the words are divided equally into *good* and *bad* words, and the users are divided equally into *original* and *shifted* users. *Original* users indicate positive and zero reward via *good* and *bad* words respectively, while *shifted* users employ the exact opposite communication convention[4].

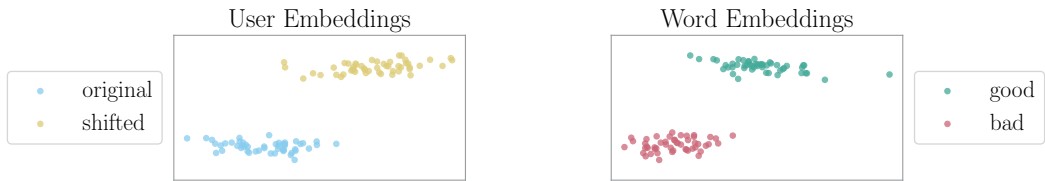

Figure 2: The proposed personalized IGL algorithm successfully disambiguates both different user communication styles and different event semantics. See Section 4.1 for details.

We simulated using the Covertype (Blackard & Dean, 1999) dataset with $M = N = 100$, and an (inverse kinematics) model class which embedded both user and word ids into a 2 dimensional space. Fig. 2 demonstrates that both the user population and the words are cleanly separated into two latent groups. Additional results showcasing the learning curves for inverse kinematics, reward and policy learning are shown in Appendix A.2 along with detailed description of model parameters.

### 4.2 FAIRNESS IN FACEBOOK NEWS RECOMMENDATION

Personalized reward learning is the key to more fair recommender systems. Previous work (Neophytou et al., 2022) suggests that inconsistent performance in recommender systems across user subgroups arises due to differences in user communication modalities. We now test this hypothesis in the setting of Facebook news recommendation. Our simulations are built on a dataset (Martinchek, 2016) of all posts by the official Facebook pages of 5 popular news outlets (ABC News, CBS News, CNN, Fox News and The New York Times) that span the political spectrum. Posts range from May to November 2016 and contain text content information, as well as logged interaction counts, which include comments and shares, as well as diverse click-based reactions (see Fig. 3).

---

[3]https://github.com/asaran/IGL-P

[4]Our motivation arises from the phenomenon of *semantic shift*, where a word's usage evolves to the point that the original and modern meaning are dramatically different (e.g. *terrific* originally meant *inspiring terror*).

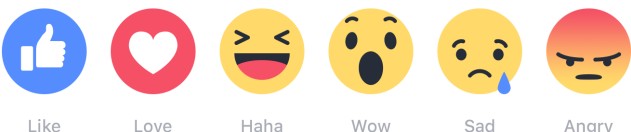

Figure 3: Facebook's click-based reactions: like, love, haha, wow, sad and angry (Meta, 2016). The reactions allow users to engage with content using diverse communication signals.

Constructing a hand-engineered reward signal using these feedbacks is difficult, and Facebook itself came under fire for utilizing reward weights that disproportionately promote toxic, low quality news. One highly criticized iteration of the reward ranking algorithm treated emoji reactions as five times more valuable than likes (Merrill & Oremus, 2021). Future iterations of the ranking algorithm promoted comments, in an attempt to bolster "meaningful social interactions" (Hagey & Jeff Horwitz, 2021). Our experiments evaluate the performance of CB algorithms (Li et al., 2010) using these two reward functions, referring to them as `CB-emoji` and `CB-comment`.

We model the news recommendation problem as a 3 latent reward state problem, with readers of different news outlets as different contexts. Given a curated selection of posts, the goal of the learner is to select the best article to show to the reader. The learner can leverage action features including the post type (link, video, photo, status or event) as well as embeddings of the text content that were generated using pre-trained transformers (Reimers & Gurevych, 2019). User feedback is drawn from a fixed probability distribution (unknown to the algorithms) that depends on the user type and latent reward of the chosen action. As an approximation of the true latent reward signal, we use low dimensional embeddings of the different news outlets combined with aggregate statistics from the post feedback to categorize whether the users had a positive ($r = 1$), neutral ($r = 0$) or negative experience ($r = -1$) with the post. This categorization is not available to the evaluated algorithms. Finally, `IGL-P(3)` uses the angry reaction as a negative oracle.

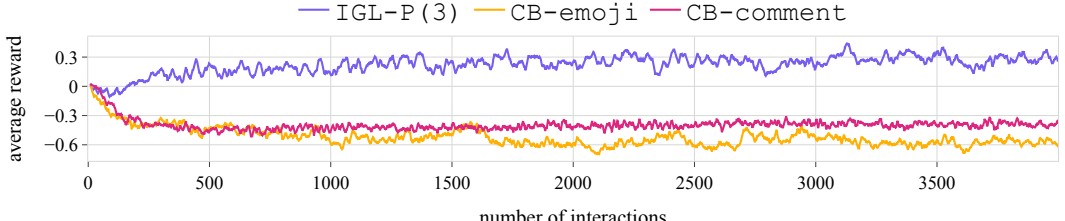

Figure 4: Of the three algorithms, only `IGL-P(3)` consistently displays posts that users enjoy.

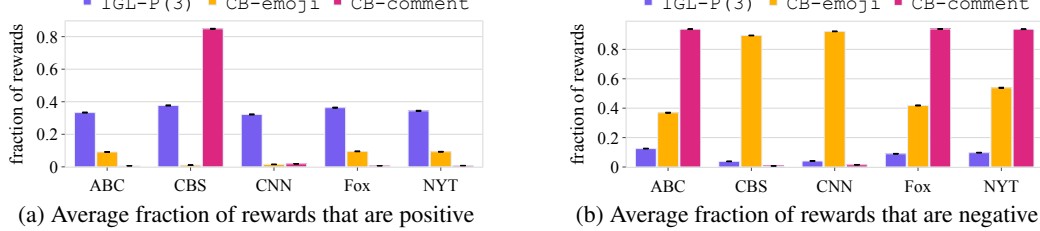

(a) Average fraction of rewards that are positive    (b) Average fraction of rewards that are negative

Figure 5: `IGL-P(3)` uses personalized reward learning to achieve fair news recommendations across diverse reader bases. CB policies based on those used by Facebook perform inconsistently, with subsets of users receiving both fewer high quality and more low quality recommendations.

Figs. 4 and 5 shows the results of our online news recommendation experiments. While the performance of both `CB-emoji` and `CB-comment` varies significantly across the different reader groups, `IGL-P(3)` provides fair performance for both positive and negative rewards. Although the `CB-comment` objective that was introduced to decrease low quality news succeeded for CBS readers, it significantly *increased* dissatisfaction among 3 of the 5 reader types (ABC News, Fox News and The New York Times). Additional details of this experiment, including model parameters and learning curves are available in Appendix A.3.

### 4.3 Production Results

Our production setting is a real world image recommendation system that serves hundreds of millions of users. In our recommendation system interface, users provide feedback in the form of clicks, likes, dislikes or no feedback. All four signals are mutually exclusive and the user only provides one feedback after each interaction. For these experiments, we use data that spans millions of interactions. The production baseline is a contextual bandit algorithm with a hand-engineered multi-task reward function which dominates approaches that only use click feedback[5]. Any improvements over the production policy imply improvement over any bandit algorithm optimizing for click feedback.

We implement `IGL-P(2)` and `IGL-P(3)` and report the performance as relative lift metrics over the production baseline. Unlike the simulation setting, we no longer have access to the user's latent reward after each interaction. As a result, we evaluate IGL by comparing all feedback signals. An increase in both clicks and likes, and a decrease in dislikes, are considered desirable outcomes. Table 1 shows the results of our empirical study. `IGL-P(2)` exhibits an inability to avoid extreme *negative* events. Although the true latent state is unknown, `IGL-P(2)` is Pareto-dominated due to an increase in dislikes. `IGL-P(3)` does not exhibit this pathology. These results indicate the utility of a 3 latent state model in real world recommendation systems. To clearly illustrate the improvement of `IGL-P(3)` and provide an additional benchmark, we also report that `IGL-P(3)` significantly outperforms a contextual bandit policy `CB-Click` using the industry standard CTR reward in Table 2.

| Algorithm | Clicks | Likes | Dislikes |
|---|---|---|---|
| `IGL-P(3)` | $[0.999, 1.067, 1.152]$ | $[0.985, 1.029, 1.054]$ | $[0.751, 1.072, 1.274]$ |
| `IGL-P(2)` | $[0.926, 1.005, 1.091]$ | $[0.914, 0.949, 0.988]$ | $[1.141, 1.337, 1.557]$ |

Table 1: Relative metrics lift over a production baseline. The production baseline uses a hand-engineered reward function which is not available to IGL algorithms. Shown are point estimates and associated bootstrap 95% confidence regions. `IGL-P(2)` erroneously increases dislikes to the detriment of other metrics. `IGL-P(3)` is equivalent to the hand-engineered baseline.

| Algorithm | Clicks | Likes | Dislikes |
|---|---|---|---|
| `IGL-P(3)` | $[1.000, 1.010, 1.020]$ | $[1.006, 1.026, 1.049]$ | $[0.890, 0.918, 0.955]$ |
| `CB-Click` | $[0.968, 0.979, 0.990]$ | $[0.935, 0.959, 0.977]$ | $[1.234, 1.311, 1.348]$ |

Table 2: Relative metrics lift over a CB policy that uses the click feedback as the reward. Point estimates and associated bootstrap 95% confidence regions are reported. `IGL-P(3)` significantly outperforms `CB-Click` at minimizing user dissatisfaction. Surprisingly, `IGL-P(3)` even outperforms `CB-Click` with respect to click feedback, providing evidence for the 3 latent state model.[6]

## 5 Related Work

**Eliminating reward engineering.** Online learning algorithms in supervised learning, contextual bandits and reinforcement learning all optimize over hand-crafted scalar rewards (Hoi et al., 2021). Previous work in areas such as inverse reinforcement learning (Arora & Doshi, 2021), imitation learning (Hussein et al., 2017) and robust reinforcement learning (Zhou et al., 2021; Panaganti et al., 2022) attempt to learn from historical data without access to the underlying true reward models. However these methods either require expert demonstration data, use hand-crafted objectives to incorporate additional human cues (Saran et al., 2020a;b; 2022) or are incapable of generalizing to different feedback signals (Chen et al., 2022), and generally need expensive compute and enormous sampling capabilities for learning. In this context, IGL (Xie et al., 2021; 2022) is a novel paradigm, where a learner's goal is to optimally *interact* with the environment with no explicit reward or

---

[5]The utility of multi-task learning for recommendation systems is well-established, e.g., Chen et al. (2021); Lu et al. (2018b); Chen et al. (2019).

[6]Data for Table 1 and Table 2 are drawn from different date ranges due to compliance limitations, so the `IGL-P(3)` performance need not match.

demonstrations to ground its policies. Our proposed approach to IGL, `IGL-P(3)`, further enables learning of personalized reward functions for different feedback signals.

**Recommender systems.** Traditional vanilla recommendation approaches can be divided into three types. Content-based approaches maintain representations for users based on their content and recommend new content with good similarity metrics for particular users (Lops et al., 2019). In contrast, collaborative filtering approaches employ user rating predictions based on historical consumed content and underlying user similarities (Schafer et al., 2007). Finally, there are hybrid approaches that combine the previous two contrasting approaches to better represent user profiles for improved recommendations (Javed et al., 2021).

Our work is a significant departure from these approaches, in that *we learn representations of user's communication style* via their content interaction history for improved diverse personalized recommendations. Specifically (1) we propose a novel personalized IGL algorithm based on the inverse kinematics strategy as described in Section 3, (2) we leverage both implicit and explicit feedback signals and avoid the costly, inefficient, status quo process of reward engineering, and (3) we formulate and propose a new recommendation system based on the IGL paradigm. Unlike previous work on IGL, we propose a personalized algorithm for recommendation systems with improved reward predictor models, more practical assumptions on feedback signals, and more intricacies described in Section 2.2.1.

## 6 DISCUSSION

We evaluated the proposed personalized IGL approach (`IGL-P`) in three different settings: (1) A simulation using a supervised classification dataset shows that `IGL-P` can learn to successfully distinguish between different communication modalities; (2) A simulation for online news recommendation based on real data from Facebook users shows that `IGL-P` leverages insights about different communication modalities to learn better policies and achieve fairness with consistent performance among diverse user groups; (3) A real-world experiment deployed in an image recommendation product showcases that the proposed method outperforms the hand-engineered reward baseline, and succeeds in a practical application.

Our work assumes that users may communicate in different ways, but a given user expresses (dis)satisfaction or indifference to all content in the same way. This assumption was critical to deriving the inverse kinematics approach, but in practice user feedback can also depend upon content (Freeman et al., 2020). IGL with arbitrary joint content-action dependence of feedback is intractable, but plausibly there exists a tractable IGL setting with a constrained joint content-action dependence. We are very much interested in exploring this as part of future work. Additionally, although we established the utility of a three state model over a two state model in our experiments, perhaps more than three states is necessary for more complex recommendation scenarios.

Although we consider the application of recommender systems, personalized reward learning can benefit any application suffering from a one-size-fits-all approach. One candidate domain is adaptive brain-computer and human-computer interfaces (Xie et al., 2022). An example is a self-calibrating eye tracker where people's response to erroneous agent control is idiosyncratic (Gao et al., 2021; Zhang et al., 2020; Saran et al., 2018). Other domains include the detection of underlying medical conditions and development of new treatments and interventions. Systematic misdiagnoses in subpopulations with symptoms different than those of white men are tragically standard in US medicine (Carter), with early stroke signs overlooked in women, minorities and young patients (Newman-Toker et al., 2014), underdetected multiple sclerosis in the Black community (Okai et al., 2022), and the widespread misclassification of autism in women (Supekar & Menon, 2015). `IGL-P` can empower medical practitioners to address systematic inequality by tuning diagnostic tools via personalization and moving past a patient-oblivious diagnostic approach. Other possible applications that can achieve improved fairness by using `IGL-P` include automated resume screening (Parasurama & Sedoc, 2022) and consumer lending (Dobbie et al., 2021).

## 7 ACKNOWLEDGEMENTS

The authors thank John Langford for the valuable discussions and helpful feedback. This material is based upon work supported by the National Science Foundation under Grant No. 1650114.

ETHICS STATEMENT

In our paper, we use two real-world interaction datasets. The first is a publicly available dataset of interactions with public Facebook news pages. All interactions are anonymous and the identities of users are excluded from the dataset, preserving their privacy. Similarly, our production data does not include user information and was used with the consent and permission of all relevant parties.

REPRODUCIBILITY STATEMENT

We have taken considerable measures to ensure the results are as reproducible as possible. We have provided the code to replicate our experiment results (except for the production results in Section 4.3) as part of the supplementary material. The code will be made publicly available at {url redacted}. We used publicly available datasets for our simulated experiments in Section 4.1 (Blackard & Dean, 1999) and Section 4.2 (Martinchek, 2016). The experiment code for Section 4.1, when executed, will automatically download the dataset. The dataset for Section 4.2 is included as part of our supplementary material. Finally, our supplementary material also includes a conda environment file to help future researchers recreate our development environment on their machines when running the experiments.

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

# A    APPENDIX

## A.1    THE 3 STATE MODEL FOR RECOMMENDER SYSTEMS

Here we provide background on the 3 latent state model for recommendation systems from Maghakian et al. (2022). In particular, we demonstrate that the traditional 2 latent state reward model used by prior IGL methods (Xie et al., 2021; 2022) is not sufficient for the recommender system scenario. We further illustrate that a 3 latent reward state model can overcome the limitations of the two latent state model, and achieve personalized reward learning for recommendations.

**Simulator Design.** Before the start of each experiment, user profiles with fixed latent rewards for each action are generated. Users are also assigned predetermined communication styles, with fixed probabilities of emitting a given signal conditioned on the latent reward. The available feedback includes a mix of explicit (likes, dislikes) and implicit (clicks, skips, none) signals. Despite receiving no human input on the assumed meaning of the implicit signals, IGL can determine which feedback are associated with which latent state. In addition to policy optimization, IGL can also be a tool for automated feature discovery. To reveal the qualitative properties of the approach, the simulated probabilities for observing a particular feedback given the reward are chosen so that they can be perfectly decoded, i.e., each feedback has a nonzero emission probability in exactly one latent reward state. Although production data does not obey this constraint (e.g., nonzero accidental feedback emissions), the not perfectly decodable setting is a topic for future work.

**IGL-P(2).** We implement Algorithm 1 for 2 latent states as `IGL-P(2)`. Our experiment shows the following two results about `IGL-P(2)`: (i) it is able to succeed in the scenario when there are 2 underlying latent rewards and (ii) it can no longer do so when there are 3 latent states. Fig. 6 shows the simulator setup used, where clicks and likes are used to communicate satisfaction, and dislikes, skips and no feedback (none) convey (active or passive) dissatisfaction.

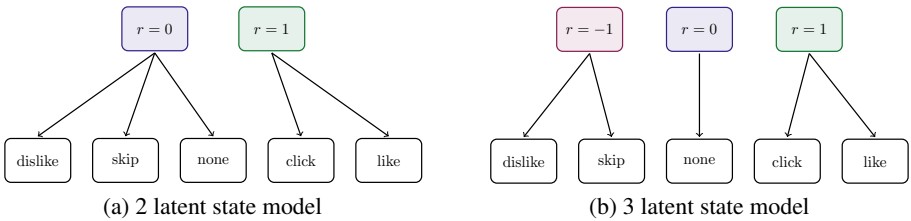

(a) 2 latent state model                          (b) 3 latent state model

Figure 6: Simulator settings for 2 state and 3 state latent model. In Fig. 6a, $r = 0$ corresponds to anything other than the user actively enjoying the content, whereas in Fig. 6b, lack of user enjoyment is split into indifference and active dissatisfaction.

Fig. 7 shows the distribution of rewards for `IGL-P(2)` as a function of the number of iterations, for both the 2 and 3 latent state model. When there are only 2 latent rewards, `IGL-P(2)` consistently improves; however for 3 latent states, `IGL-P(2)` oscillates between $r = 1$ and $r = -1$, resulting in much lower average user satisfaction. The empirical results demonstrate that although `IGL-P(2)` can successfully identify and maximize the rare feedbacks it encounters, it is unable to distinguish between satisfied and dissatisfied users.

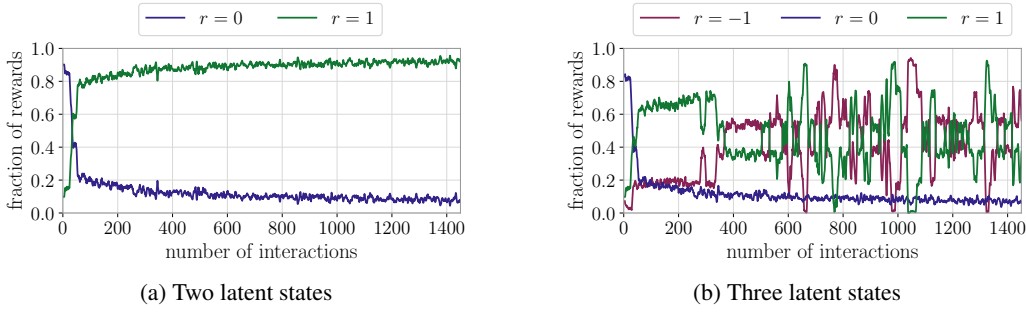

(a) Two latent states                          (b) Three latent states

Figure 7: Although `IGL-P(2)` is successful with the 2 state simulator, it fails on the 3 state simulator and oscillates between attempting to maximize $r = 1$ and $r = -1$.

**IGL-P(3): Personalized Reward Learning for Recommendations.** Since `IGL-P(2)` is not sufficient for the recommendation system setting, we now explore the performance of `IGL-P(3)`. Using the same simulator as Fig. 6b, we evaluated `IGL-P(3)`. Fig. 8a demonstrates the distribution of the rewards over the course of the experiment. `IGL-P(3)` quickly converged, and because of the partial negative feedback for dislikes, never attempted to maximize the $r = -1$ state. Even though users used the ambiguous skip signal to express dissatisfaction 80% of the time, `IGL-P(3)` was still able to learn user preferences.

In order for `IGL-P(3)` to succeed, the algorithm requires direct grounding from the dislike signal. We next examined how `IGL-P(3)` is impacted by increased or decreased presence of user dislikes. Fig. 8b was generated by varying the probability $p$ of users emitting dislikes given $r = -1$, and then averaging over 10 experiments for each choice of $p$. While lower dislike emission probabilities are associated with slower convergence, `IGL-P(3)` is able to overcome the increase in unlabeled feedback and learn to associate the skip signal with user dissatisfaction. Once the feedback decoding stabilizes, regardless of the dislike emission probability, `IGL-P(3)` enjoys strong performance for the remainder of the experiment.

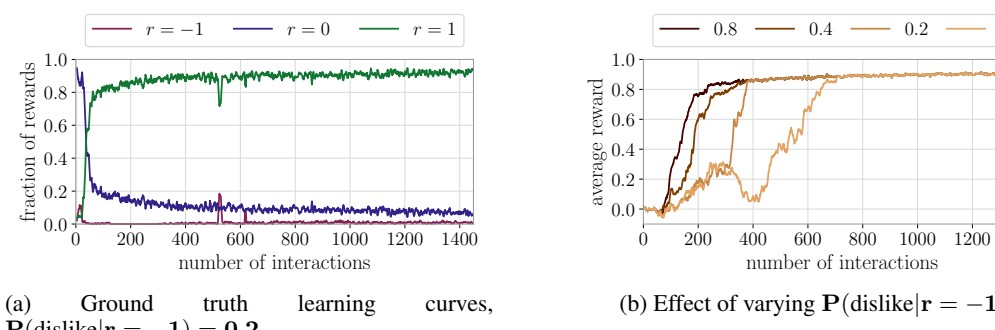

(a) Ground truth learning curves, $\mathbf{P}(\text{dislike}|\mathbf{r} = -\mathbf{1}) = \mathbf{0.2}$.

(b) Effect of varying $\mathbf{P}(\text{dislike}|\mathbf{r} = -\mathbf{1})$.

Figure 8: Performance of `IGL-P(3)` in simulated environment. In Fig. 8a, `IGL-P(3)` successfully maximizes user satisfaction while minimizing dissatisfaction. Fig. 8b demonstrates how `IGL-P(3)` is robust to varying the frequency of partial information received, although more data is needed for convergence when "definitely bad" events are less frequent.

## A.2 ADDITIONAL RESULTS FOR THE COVERTYPE IGL SIMULATION

**Model Parameters For The Covertype IGL Simulation.** For the Covertype IGL simulation we trained both a CB and IK model (as shown in Algorithm 1). We use online multiclass learning to implement the IK model, where the classes are all the possible actions and the label is the action played. Both CB and IK are linear logistic regression models implemented in PyTorch, trained using the cross-entropy loss. Both models used Adam to update their weights with a learning rate of $2.5e^{-3}$. The first layer of weights for both models had $512$ weights that were initialized using a Cauchy distribution with $\sigma = 0.2$. The learning curves are shown in Figure 9b.

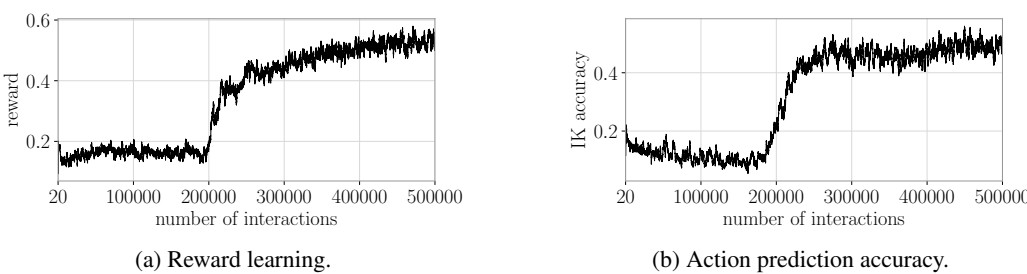

(a) Reward learning.

(b) Action prediction accuracy.

Figure 9: Learning curves for the Covertype simulated experiment in Sec. 4.1. Figure 9b shows the accuracy with which the inverse kinematics model recovered the actions.

## A.3 ADDITIONAL RESULTS FOR FACEBOOK NEWS SIMULATION

**Model Parameters For The Facebook News Simulation.** For the Facebook news simulation there were 2 CB models and an IK model. All models used Adam to update their weights with a learning rate of $10^{-3}$, batch size 100, and cross-entropy loss function. The reward learning curves are shown in Figures 10,11 and 12. Further details about implementation are available in provided code[7].

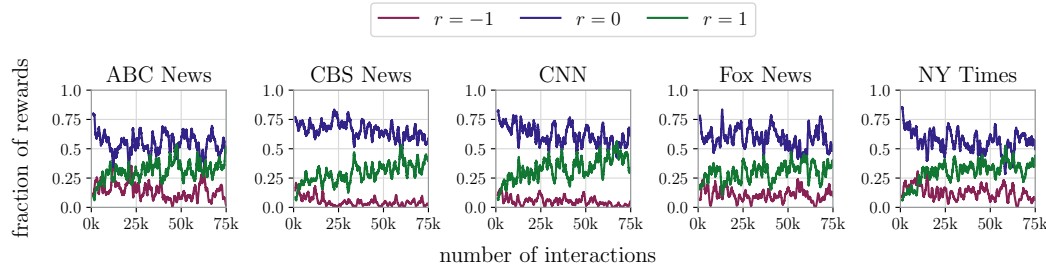

Figure 10: Reward learning curves for `IGL-P(3)` in the Facebook news recommendation experiment. Over time, `IGL-P(3)` learns the difference between the three states and tries to minimize $r = 0$ and $r = -1$ while maximizing $r = 1$ across all contexts.

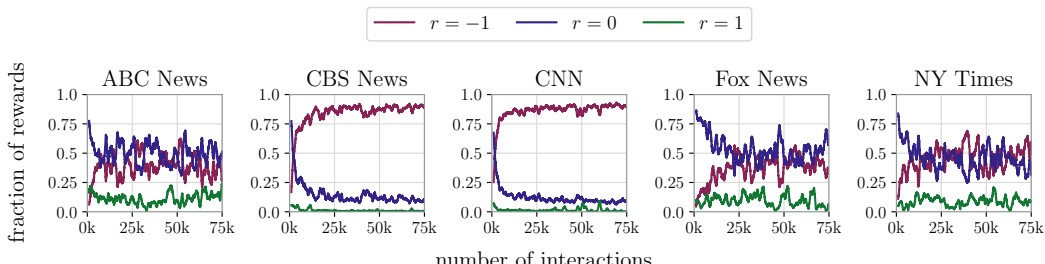

Figure 11: Reward learning curves for `CB-emoji` in the Facebook news recommendation experiment. Two of the five reader types (CBS News and CNN) communicate dissatisfaction using many of the emoji feedback. As a result, `CB-emoji` ultimately maximizes $r = -1$ for those readers.

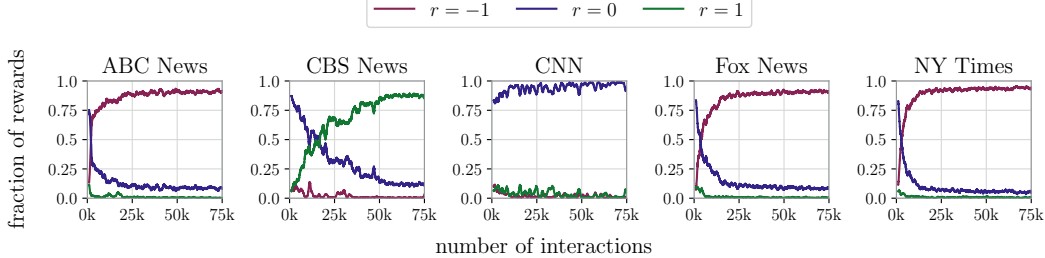

Figure 12: Reward learning curves for `CB-comment` in the Facebook news recommendation experiment. Although `CB-comment` performs successfully with CBS News readers, it maximizes the unhappiness of ABC News, Fox News and The New York Times readers.

---

[7]https://github.com/asaran/IGL-P

