# OpenReview forum: "Personalized Reward Learning with Interaction-Grounded Learning (IGL)"
_ICLR.cc/2023/Conference — ICLR 2023 poster_

### Official Review · Reviewer_4RJN · 2022-10-25

**Confidence:** 3
**Correctness:** 3
**Technical Novelty And Significance:** 3
**Empirical Novelty And Significance:** 4
**Recommendation:** 6

**Clarity, Quality, Novelty And Reproducibility:**

The quality and originality of the work is high, while the writing can be further improved for clarity. Especially, Algorithm 1 is not very informative. Instead of using ‘.predict’ and ‘.learn’, it would be clearer if the author provide the exact loss function and steps used to train KI and Policy, as for RecSys audience who are less familiar with KI it is not obvious how KI is learned.

**Strength And Weaknesses:**

Strength:

- The authors find a novel application of IGL paradigm on personalized RecSys and propose key modifications that adapt to user modeling use case. The resulting model solves an important problem in RecSys where we usually lack explicit feedback on the objective that need to be optimized, and implicit feedbacks are sensitive to user subgroups such that ad-hoc reward engineering is undesirable.
- The application of Inverse Kinematics on IGL problem is novel, and it is technically non-trivial.
- The introduction of 3-state reward decoder is non-trivial and interesting.
- The result on synthetic feedback scenario shows improved fairness, and the algorithm is comparable to SOTA production candidate with hand-crafted reward engineering.

Weakness:

Despite the novelty and contributions, there are some ambiguities that may need further clarification

- When and how is the DN(…) function learned? It seems PU-learning is used where only negative labels are partially provided, and the labeling probability is somehow known. But in the empirical experiments it is unclear how these set of information is obtained. Moreover, it is non-trivial to estimate labeling probability in real-world, and there is not much explanation on how this hyperparameter is chosen and how it impact the performance.
- The synthetic experiments with Facebook and Covertype datasets are designed such that feedbacks are heavily dependent on user type, making it in favor of the proposed algorithm. And only simple CB are used as baselines for comparison. The real-production result is comparable to reward-engineering, while the significance of the lift is not clear as the confidence lower bound <1 for likes and there is a positive lift in dislike. Comparison to action-inclusive IGL in Xie et al 2022 is also lacking, making it less convincing whether the user dependence assumption is the main reason for the lift.
- The authors propose different assumption and learning algorithm than vanilla IGL. Yet no theoretical analysis is provided to prove whether the new independence assumption and the learning algorithm converges to the optimal solution. The enhancement on fairness is also supported with only empirical evidence, thus might limit the impact of the work.

Minor issues:

- There might be a typo in the last line of equation (2) where the prior on numerator is `p(a,x)` whereas in equation(3) it is `p(a|x)`.
- Missing figure for learning curves for IK, reward and policy learning.


**Summary Of The Paper:**

In this paper, the authors propose a new algorithm for Personalized RecSys under Interaction Grounded Learning (IGL) paradigm where a personalized policy is learned to maximum unobservable user satisfaction with only implicit feedbacks. The authors alter the independence assumption used in vanilla IGL (Xie et al 2021/2022) to enable personalized feedback that depends on the context. They also increase the reward state from 2 to 3 to capture non-extreme (neutral) user state. Based on the modified assumptions, the authors propose a new IGL algorithm with Inverse Kinematics (different from contrastive learning algorithm proposed in original IGL in 2022) and a novel reward decoder that handles three states.  Empirical results are shown with both simulation and real-world News recommendation dataset, where better fairness is achieved when feedbacks are synthetically designed to be user dependent. They also AB test in production setting and is comparable with contextual bandit with reward engineering.

**Summary Of The Review:**

The paper proposes novel solution for personalized reward learning with implicit feedback in IGL paradigm, and come up with important modification of original IGL (Xie et al 2022). Despite limited theoretical result and baselines, the contribution is potentially significant to the community.

---

> ### Author Response · Authors · 2022-11-17
> **Response to Reviewer 4RJN (2/2)**
>
> - > The authors propose different assumption and learning algorithm than vanilla IGL. Yet no theoretical analysis is provided to prove whether the new independence assumption and the learning algorithm converges to the optimal solution. The enhancement on fairness is also supported with only empirical evidence, thus might limit the impact of the work.
>
>     We agree with the reviewer that we make different assumptions compared to prior work on IGL [1] [2], and our proposed work in its current form assumes access to infinite amounts of data for strong convergence guarantees. While it is impractical to have access to infinite amounts of data in practice, we note that our work highlights the strong empirical results even with the finite data as part of our experiments. Theoretically characterising the performance of the proposed approach IGL-P(3) via finite sample complexity analysis is an ongoing part of our future work.
>
>     [1] T. Xie , J. Langford, P. Mineiro, and I. Momennejad. "Interaction-Grounded Learning." ICML 2021.
>     [2] T. Xie, A. Saran, D.J. Foster, L. Molu, I. Momennejad, N. Jiang, P. Mineiro, J. Langford. "Interaction-Grounded Learning with Action-Inclusive Feedback." NeurIPS 2022.
>
> - > There might be a typo in the last line of equation (2) where the prior on numerator is p(a,x) whereas in equation(3) it is p(a|x).
>
>     We thank the reviewer for pointing out this typing mistake. We have made the correction in the main draft (highlighted in magenta).
>
> - > Missing figure for learning curves for IK, reward and policy learning.
>
>     We thank the reviewer for pointing out the reference to the missing figures. We’ve appropriately updated the reference in the main text for learning curves of the Covertype simulation as well as included the learning curves for the Facebook news recommendation experiment.
>
> - > …the writing can be further improved for clarity. Especially, Algorithm 1 is not very informative. Instead of using ‘.predict’ and ‘.learn’, it would be clearer if the author provide the exact loss function and steps used to train KI and Policy, as for RecSys audience who are less familiar with KI it is not obvious how KI is learned.
>
>     We thank the reviewer for highlighting this lack of clarity. To clarify further, we use online multiclass learning to implement the IK model, where the classes are all the possible actions and the label is the action played. Both IK and CB policy models are implemented as linear logistic regression in PyTorch, trained using the cross entropy loss and a standard optimizer (Adam) is used to learn these models online.
>
>     In Algorithm 1, the ‘IK.predict’ function refers to the inference of the IK model predicting the action that was played, ‘\pi.predict’ refers to the inference of the CB policy model which outputs a probability distribution over all actions. The ‘.learn’ function calls refer to the optimization update step for the IK and CB models. Please let us know if further clarification is needed.

---

> ### Author Response · Authors · 2022-11-17
> **Response to Reviewer 4RJN (1/2)**
>
> We thank the reviewer for their thoughtful comments on our paper.  We are encouraged by the fact that they found our proposed method novel, interesting, and useful. We address the reviewer’s comments below and have updated our paper accordingly (changes in magenta). Please let us know if further clarification is needed.
>
> - > When and how is the DN(…) function learned? It seems PU-learning is used where only negative labels are partially provided, and the labeling probability is somehow known. But in the empirical experiments it is unclear how these set of information is obtained. Moreover, it is non-trivial to estimate labeling probability in real-world, and there is not much explanation on how this hyperparameter is chosen and how it impact the performance.
>
>     We assume partial reward information is available via a definitely negative (DN) function. The DN function is implemented as a pytorch model with a single output, where a label of  -1 represents a definitely negative event whereas a label of 0 represents an event that is not definitely negative (Sec. 3, extreme event disambiguation subsection, Equation 4).
>
>     Yes, the label probability is a prior for the PU learning approach and would need to be tuned like a hyperparameter for the application, based on preferences over the observed metrics that are available offline. Without additional assumptions such as a labeling prior,  PU learning is misspecified.  However, in practice, we find that the algorithm is not sensitive to the labeling probability in a broad range (0.5 to 0.99). We include some additional results to highlight this observation:
>
>     Sensitivity of prior probability:
>
>     | Algorithm              | Clicks                | Likes                 | Dislikes              |
>     |------------------------|-----------------------|-----------------------|-----------------------|
>     | IGL-P(3) (Prior = 0.5) | [0.990, 1.003, 1.016] | [1.027, 1.065, 1.104] | [0.898, 0.945, 1.002] |
>     | IGL-P(3) (Prior = 0.9) | [0.998, 1.014, 1.031] | [1.016, 1.052, 1.090] | [0.873, 0.934, 0.984] |
>
>
>     There is no statistical difference between the performance of IGL-P(3) for prior values of 0.5 and 0.9. The confidence intervals completely overlap. In our production scenario, dislikes represent the definitely negative event, hence a prior close to 1 works better.
>
>     We note that we will release the code for all experiments except the production scenario (also included as part of the appendix).
>
> - > The synthetic experiments with Facebook and Covertype datasets are designed such that feedbacks are heavily dependent on user type, making it in favor of the proposed algorithm. And only simple CB are used as baselines for comparison. The real-production result is comparable to reward-engineering, while the significance of the lift is not clear as the confidence lower bound <1 for likes and there is a positive lift in dislike. Comparison to action-inclusive IGL in Xie et al 2022 is also lacking, making it less convincing whether the user dependence assumption is the main reason for the lift.
>
>     We claim that even matching the performance of the manually designed reward function is a notable contribution, and eliminates the need for hand-crafting rewards. In our production setting, the current reward function is nontrivial and is the result of years of trial and error tuning by product engineers and data scientists. The cost of reward engineering is visible in many industrial recommender systems, e.g. Facebook’s many iterations of emoji reaction weights. Furthermore, the eventual design of a “good” reward does not mean that it will remain “good” as system interfaces and users evolve. As a result, the process of manual reward engineering becomes a very costly, ongoing expense.
>
>     We do not compare to action-inclusive IGL due to the fact that in our dataset, conditioning on any given action does not result in a statistically different feedback profile. Since the feedback does not depend on the action, action-inclusive IGL does not provide any additional improvements. However, there was strong statistical evidence of feedback dependence on user demographic. Since action-inclusive IGL does not adapt to different users, it would be unable to handle this setting and perform poorly. Although our production setting did not demonstrate action dependence, we are very much interested in exploring the challenging setting of constrained joint context-action dependence on feedback in future work.

---

### Official Review · Reviewer_iWcq · 2022-10-27

**Confidence:** 3
**Correctness:** 3
**Technical Novelty And Significance:** 3
**Empirical Novelty And Significance:** 3
**Recommendation:** 6

**Clarity, Quality, Novelty And Reproducibility:**

Please refer to my comments above for clarity issues.

I think the idea of IGL for recommendation is novel and reasonable. The proposed method also solves two domain-specific challenges, though left with many others to be further tackled.

The authors also provide the source code for reproducibility check.


**Strength And Weaknesses:**

Strengths:
1. This paper tackles an important research problem, optimizing latent user satisfaction in recommender systems, through an interesting angle, i.e., learning personalized reward functions.
2. The proposed IGL method is reasonable and capable of solving domain-specific challenges.
3. The proposed method is rigorously derived in Section 3.

Weaknesses:
1. Background on contextual bandit and interaction grounded learning should be strengthened.
2. The authors claimed that this method could handle both implicit and explicit feedback, while only implicit feedback is considered in experiments.
3. The empirical performance improvements seem to be barely marginal, and only three types of contexts are considered in the facebook news recommendation scenario. What is the number of contexts on production datasets?
4. Many details of experiments and implementations are missing, like the definition of metrics, scale of production dataset, model parameters, etc.
5. The paper is generally well-written, but its clarity can be further improved. For example, the meaning of $a$ / $r$ in recommender systems should be clearly defined.


**Summary Of The Paper:**

This paper proposes a personalized reward learning method for recommender systems. The authors apply the recent Interaction Grounded Learning (IGL) paradigm to address the challenge of learning representations of diverse user communication modalities. The proposed personalized IGL is designed for context-dependent feedback, with inverse kinematics as an IGL objective and the capability of modeling more than two latent states. Both simulations and experiments on real production data have demonstrated the proposed IGL is able to address two typical challenges for modern online recommender systems, i.e., feedback-reward dependence assumption and the number of latent reward states.

**Summary Of The Review:**

Given the listed strengths and weaknesses, I think this is a borderline paper.

## After rebuttal
The authors' response has somewhat alleviated my concerns. Thus I decide to raise my score from 5 to 6.

---

> ### Author Response · Authors · 2022-11-17
> **Response to Reviewer iWcq**
>
> We thank the reviewer for their thoughtful comments on our paper.  We are encouraged by the fact that they found our proposed approach to be novel and rigorous. We address their  concerns below and have updated the paper accordingly (changes in magenta). Please let us know if further clarification is needed.
>
> - > The authors claimed that this method could handle both implicit and explicit feedback, while only implicit feedback is considered in experiments.
>
>     We incorporate explicit feedback in the form of the oracle grounding signal for disambiguating extreme positive events and extreme negative events. For our experiments, we use explicit negative signals such as dislikes or angry emojis as the negative oracle, but in principle any explicit feedback available to the learner can be incorporated.
>
> - > The empirical performance improvements seem to be barely marginal, and only three types of contexts are considered in the facebook news recommendation scenario. What is the number of contexts on production datasets?
>
>     **IGL-P eliminates the need for hand-engineering of reward functions.**
>     The performance of IGL-P on our production system is impressive given that our baseline state-of-the-art competitor policy uses a hand-engineered reward function that was manually designed and tuned by product engineers over the span of years, and the policy itself was trained on years of data, in comparison with IGL-P which was cold-started and trained on at most 10 days of data.
>
>     To give some sense of the scale of our production system, we have thousands of high-dimensional actions, tens of millions of unique user interactions, and hundreds of different contexts. Exact values are not provided due to proprietary concerns and data compliance requirements.
>
> - > Many details of experiments and implementations are missing, like the definition of metrics, scale of production dataset, model parameters, etc.
>
>     We have updated the appendix to include the implementation details of our experiments.
>
> - > …clarity can be further improved. For example, the meaning of a/ r in recommender systems should be clearly defined.
>
>     We have updated the paper to clearly define this information in Section 2.2.1.

---

### Official Review · Reviewer_7wvr · 2022-10-31

**Confidence:** 4
**Correctness:** 3
**Technical Novelty And Significance:** 3
**Empirical Novelty And Significance:** 3
**Recommendation:** 6

**Clarity, Quality, Novelty And Reproducibility:**

Overall the paper is well written, and there is some degree of novelty in that IGL is being applied to the recommendation problem and in particular to mapping implicit signals to rewards. The treatment of the problem and the derivation of the mapping to rewards could be clearer as it is somewhat difficult to follow. It would be helpful if more of an intuition is provided at every step.
The author seem to be willing to publish the code and most datasets from the experimental section which would provide adequate reproducibility of a big part of the experimental section.


**Strength And Weaknesses:**

The paper is well written and well motivated, in particular, the issue of extracting reward out of implicit signals in recommendations is an open question. The paper also includes results from a production system that is not typically the case in this type of work. Moreover, the idea of applying IGL principles in the area of recommender systems is to the best of my knowledge novel and justified well.
The experimental section is somewhat restricted to results from the live experiment and there is no clear comparison to e.g. bandit algorithms with a simpler implicit signal to reward mapping hence it is unclear how much of an advantage the method hold compared to potentially simpler alternatives. It is also somewhat unclear against what baseline the method is tested in the live experiments.
One somewhat negative point is that this work is relatively narrow even in the space of recommender systems and might interest only a small subset of the ICLR audience, a more IR/recommender systems conference might be a more appropriate venue.

**Summary Of The Paper:**

The paper posits that reward signals from users are not necessarily clear and cannot be easily attributed to genuine preference but might be the result of various biases. In particular implicit signals (watches/clicks) do not map directly to user satisfaction. Moreover, different users can have different ways of signaling preferences. The authors propose Interaction Grounded Learning or personalized
reward learning (IGL). IGL is a learning paradigm where a learner optimizes for unobservable
rewards by interacting with the environment and associating observable feedback with the true latent reward.
The adaptations to the case of recommendation of IGL are presented in the paper and the final formulation of the algorithm involves a mapping between conditional probabilities and estimated potential rewards. On that basis, a Bandit-type algorithm is trained and experiments are conducted on artificial offline data and real news recommendation data with positive results.



**Summary Of The Review:**

Overall an interesting work with a novel treatment of a problem in the area of recommender systems.
I'm a bit unsure about the impact of the work in a general ML conference.

---

> ### Author Response · Authors · 2022-11-17
> **Response to Reviewer 7wvr (2/2)**
>
> - > The treatment of the problem and the derivation of the mapping to rewards could be clearer as it is somewhat difficult to follow. It would be helpful if more of an intuition is provided at every step.
>
>     We thank the reviewer for the feedback and have updated the paper to provide some more intuition for the use of IK in our approach (Sec 2.2.1 and Sec 3). We note that we use online multiclass learning to implement the IK model, where the classes are all the possible actions and the label is the action played. Both the CB and IK models are implemented as linear logistic regression models, trained online with a cross-entropy loss and a standard pytorch optimizer (Adam).
>
>     In Algorithm 1, the ‘IK.predict’ function refers to the inference of the IK model predicting the action that was played. The ‘IK.learn’ function call refers to the optimization update step for the IK model. The output of the IK.predict function and the formulation in Equation 4, guides the learning of the CB model.
>
>     Please let us know if further clarification is needed.

---

> ### Author Response · Authors · 2022-11-17
> **Response to Reviewer 7wvr (1/2)**
>
> We thank the reviewer for their thoughtful comments on our paper.  We are encouraged by the fact that they found our work novel in the sense of IGL’s applicability to recommender systems. We note that the proposed approach IGL-P is also novel in the following ways: (1) the first IGL strategy for context-dependent feedback, (2) the first use of inverse kinematics as an IGL objective, and (3) the first IGL strategy for more than two latent states. We address the reviewer’s concerns below and we have updated our paper accordingly (changes highlighted in magenta). Please let us know if further clarification is needed.
>
> - > The experimental section is somewhat restricted to results from the live experiment and there is no clear comparison to e.g. bandit algorithms with a simpler implicit signal to reward mapping hence it is unclear how much of an advantage the method holds compared to potentially simpler alternatives. It is also somewhat unclear against what baseline the method is tested in the live experiments.
>
>     **IGL-P eliminates the need for hand-engineering of reward functions.**
>     The baseline policy is trained using a hand-engineered reward function that is non-intuitive and the result of trial and error research over many years by product engineers and data scientists. Furthermore, the baseline policy was trained on years of data compared to our IGL-P approach which was cold-started and trained on at most 10 days of data.
>
>     We have also updated the paper to include results comparing IGL-P with a simple bandit algorithm using click feedback (Table 2 in Sec. 4.3).
>
>     | Algorithm | Clicks                 | Likes                 | Dislikes              |
>     |-----------|------------------------|-----------------------|-----------------------|
>     | IGL-P(3)  |  [1.000, 1.010, 1.020] | [1.006, 1.026, 1.049] | [0.890, 0.918, 0.955] |
>     | CB-Click  |  [0.968, 0.979, 0.990] | [0.935, 0.959, 0.977] | [1.234, 1.311, 1.348] |
>
>    The table above shows relative metrics lift over a CB policy that uses the click feedback as the reward. Point estimates and associated bootstrap 95% confidence regions are reported.  Note that data for Table 1 and Table 2 in the paper are drawn from different date ranges due to compliance limitations, so the IGL-P(3) performance need not match.
>
> - > One somewhat negative point is that this work is relatively narrow even in the space of recommender systems and might interest only a small subset of the ICLR audience, a more IR/recommender systems conference might be a more appropriate venue.
>
>     Although we demonstrate the success of IGL-P in a recommender system setting, we note that the methods can be used in many other application domains of interest to ML practitioners. Our proposed approach is a generic solution towards recovering personalised reward functions via interaction, moving away from hand-designed reward functions — which would be of interest to a broader audience of researchers at a venue such as ICLR.
>
>     As an example of broader applicability of IGL-P, one candidate domain is the detection of underlying medical conditions and development of new medical treatments and interventions. In the United States, this domain has historically suffered from inequality arising from “a deeply rooted concept that the 70kg white male is ‘normal’” and other individuals “are deviations from that model.” Although different subpopulations can experience different symptoms arising from the same underlying cause, medical practitioners have historically been trained to look for the symptoms that occur in white men. As a consequence, women are 50% more likely than men to receive an incorrect diagnosis after a heart attack. This is due in part to the fact that women are more likely to experience “uncommon” symptoms compared to men. The clear user-dependence of feedback signals in medical settings suggests the potential of applying IGL-P. Other potential applications are learned brain-computer and human-computer interfaces. One example is a self-calibrating eye tracker where people's response to erroneous agent control is idiosyncratic. We have also included these examples in Sec. 6 (shown in magenta), as well as updated the related work section to explore the literature on reward engineering and highlight our novel contributions.

---

### Author Response · Authors · 2022-12-10
**Thanks to the Reviewers and Summary of Changes in the Revision**

As the discussion period is about to end, we would like to thank all the reviewers for their attention and constructive feedback. We summarize the updates we made to the draft for consideration of the reviewers one last time:

- Interaction-Grounded Learning (IGL) is a relatively novel machine learning paradigm (Xie et al., 2021; 2022), where a learner’s goal is to optimally interact with the environment with no explicit reward or demonstrations to ground its policies. Our proposed approach **IGL-P is a novel personalized variant of IGL**: the first IGL strategy for context-dependent feedback, the first use of inverse kinematics as an IGL objective, and the first IGL strategy for more than two latent states.

- **IGL-P learns a latent representation of the reward, eliminating the need for hand-engineering of reward functions**. Although we consider the application of recommender systems, **personalized reward learning can benefit any application suffering from a one-size-fits-all approach**. We provide some examples for such applications (AI-assisted medical diagnosis, intelligent brain- and human-computer interfaces etc.) in response to reviewer 7wvr as well as the discussion section of the paper. We believe our proposed approach will be of value to a part of the ICLR community interested in **interactive machine learning** and **fairness**.

- We provide more intuitive explanations for our IK-based approach in Sec. 2 and Sec. 3.

- For the production dataset, we include additional results with a baseline in Table 2 (bandit algorithm using click feedback) and show that **IGL-P** significantly outperforms this baseline.

- We provide information about the models, optimization and hyperparameters used  as well as additional results on simulated datasets in the Appendix.

Please let us know if there are any last minute questions related to our approach, experiments, contextualization with prior work, or potential use cases of **IGL-P** that we can clarify for you.



**References**:

Tengyang Xie, John Langford, Paul Mineiro, and Ida Momennejad. "Interaction-grounded learning". In International Conference on Machine Learning, pages 11414–11423. PMLR, 2021.

Tengyang Xie, Akanksha Saran, Dylan J. Foster, Lekan Molu, Ida Momennejad, Nan Jiang, Paul Mineiro, and John Langford. "Interaction-Grounded Learning with Action-inclusive Feedback." Advances in Neural Information Processing Systems, 2022.

---

### Decision · Program_Chairs · 2023-01-20

**Decision:**

Accept: poster

**Justification For Why Not Higher Score:**

Missing details on implementations and experiments. Only compare to simple baselines. Might not have a big audience at the conference.

**Justification For Why Not Lower Score:**

Overall reviewers appreciate the application of IGL to address the challenge of lacking explicit feedback and the need of reward engineering in recommender systems.

**Metareview: Summary, Strengths And Weaknesses:**

The paper posits that users communicates their true preferences differently, thus relying on a fixed, human-designed reward function on implicit user feedbacks to come up with the underlying user preferences is sub-optimal. The authors proposed to combine the new Interaction Grounded Learning (IGL) paradigm with inverse kinematics to learn personalized/context-dependent reward function, and showed promising results in offline simulated and live experiments. Reviewers converged to a recommendation of marginally above acceptance threshold after rebuttal. Overall reviewers appreciate the application of IGL to address the challenge of lacking explicit feedback and the need of reward engineering in recommender systems, and ask for more details on the implementations and experiments.


**Note From Pc:**

if the above contains the word "oral" or "spotlight" please see: "oral" presentation means -> notable-top-5% and "spotlight" means -> notable-top-25%. As stated in our emails, we are disassociating presentation type from AC recommendations